# DETECTING MALICIOUS PDF USING CNN

## ABSTRACT

Malicious PDF files represent one of the biggest threats to computer security. To detect them, significant research has been done using handwritten signatures or machine learning based on manual feature extraction. Those approaches are both time-consuming, requires significant prior knowledge and the list of features has to be updated with each newly discovered vulnerability. In this work, we propose a novel algorithm that uses a Convolutional Neural Network (CNN) on the byte level of the file, without any handcrafted features. We show, using a data set of 130000 files, that our approach maintains a high detection rate (96%) of PDF malware and even detects new malicious files, still undetected by most antiviruses. Using automatically generated features from our CNN network, and applying a clustering algorithm, we also obtain high similarity between the antiviruses' labels and the resulting clusters.

## 1 INTRODUCTION

Malware programs are still making newspapers' headlines. They are used by criminal organizations, governments, and industry to steal money, spy or other unwanted activities. As millions of new malicious samples are discovered every day, spotting them before they harm a computer or a network remains one of the most important challenges in cybersecurity. During the last two decades, hackers kept finding new attack vectors, giving malware multiple forms. Some use the macros in Office documents while others exploit browser's vulnerabilities with javascript files. This diversity raised a need for new automated solutions.

PDF is one of the most popular types of documents. Despite the lack of awareness of the population, it also became an important attack vector for computer systems. Dozens of vulnerabilities are discovered every year on Adobe Reader, the most popular software for reading PDF files (1), allowing hackers to take control of the victim's computer. PDF malware can be segmented into three main categories: (i) exploits, (ii) phishing and (iii) misuse of PDF capabilities. Exploits operate by taking advantage of a bug in the API of a PDF reader application, which allows the attacker to execute code in the victim's computer. This is usually done via JavaScript code, embedded in the file. In phishing attacks, the PDF itself does not have any malicious behavior but attempts to convince the user to click on a malicious link. Such campaigns have been discovered recently (2) and are, by nature, much harder to identify. The last category exploits some regular functionality of PDF files such as running a command or launching a file.All those attacks can lead to devastating consequences, such as downloading a malicious executable or stealing credentials from a website.

Regardless of recent work in machine learning for malware detection, antivirus companies are still largely focusing on handwritten signatures to detect malicious PDF. This not only requires a lot of human resources but is also rarely efficient at detecting unknown variants or zero day attacks. Another popular solution is dynamic analysis by running the files in a controlled sandboxed environment. Such approaches increase significantly the chance of detecting new malware, but take much longer and requires access to a sandbox virtual machine. They also still require a human to define the detection rules according to the file behavior.

In this paper, we are using a Convolutional Neural Network (CNN) in order to detect any type of malicious PDF files. Without any preprocessing of the files, our classifier succeeds to detect 94% of the malicious samples of our test set while keeping a False Positive Rate (FPR) at 0.2%. Our classifier outperforms most of the antiviruses (AV) vendors available in the *VirusTotal* website.We also show that our network can successfully group more than 80% of the malware into different

families agreeing at 76% with the name given by the AV. Finally, we will present some examples on which we were able to detect an attack before the AV (zero-day).

To the best of our knowledge, this is the first paper using Convolutional Neural Network to classify malicious PDF files. It is also the first one that investigates the ability to automatically classify PDF malware into different families.

**Paper organization:** We first present the related researches that have been done in machine learning for detecting malicious PDF and the usage of Deep Learning applied to Malware Detection in executable files (section 2). We describe how we built our data set in Section 3, and describe our model in Section 4. We show our results on the data sets in Section 5. We investigate the capability of our network to differentiate between malware types in Section 6. Our conclusion is in Section 7.

## 2    RELATED WORK

### 2.1    MACHINE LEARNING FOR DETECTING MALICIOUS PDF

Signature-based detection used to be the standard in cybersecurity, and it is the preferred solution where researchers use signatures to identify malicious PDF (13). However, with the fast increase of threats, the work required by handwritten rules increased significantly, and machine learning has been extensively used in the last decade to enrich detection capabilities.

Maiorca, Giacinto and Corona (9) generate features based on keywords (tags) found in the PDF files. They choose the keywords by clustering appearances in benign and malicious sets. They obtain high classification results (99.6% of detection), but their approach mainly focuses on specific types of malware, a subset that does not include exploits and phishing PDF. Munson and Cross (4) define a list of features extracted from both static and dynamic analysis of the PDF files, and train a decision tree classifier, but they used a very small dataset of 87 malicious samples. Tzermias et al. (17) combine static and dynamic analysis, focusing on specific vulnerabilities, and managed to detect 89% of the malware in their dataset. However, it takes 1.5s for their algorithm to run on a single PDF file and requires the usage of a VM.

Stavrou and Smutz (15) focus their feature extraction on metadata and structure of the documents. They end up with 202 manually selected features that are used to build a Random Forest classifier and obtain a detection of more than 99% and a FPR lower than 0.5%. Srndic and Laskov (16) use an open source PDF parser and retrieve the tree-like structure of the file which they use as a feature. They then train some Decision Trees and a SVM on several datasets and manage to detect 95% of new files, keeping an FPR around 0.1%. More recently, Zhang (18) is using Multi-Layer Perceptron (MLP) on 48 manually selected features, and achieves a better detection than 8 antiviruses in the market, obtaining a detection rate of around 95% on their datasets.

We emphasise, that all the previous approaches require prior domain knowledge as they are based on manual feature selection. As far as we know, our work is the first to automate the feature extraction of PDF files.

### 2.2    DEEP LEARNING IN MALWARE DETECTION

In the last years, there have been several efforts in using Deep Learning for detecting malware in executable files (type `exe`). David and Netanyahu, in DeepSign (5), are using a Deep Belief Network with Denoising Autoencoders to automatically generate signatures based on executables' behavior. To do so, they run the files on a sandbox, extract the API calls and create 5000 one-hot encoded features out of them. The signatures consist of 30 features generated by the network that are used in the classifier and achieves a 98.6% of accuracy on their dataset. Pascanu et al. (11), starting from the API calls, are generating an embedding of the malware behavior using Echo State Network and Recurrent Neural Network. They train the network to predict the next API call and use the last hidden state as a feature for a classifier. They obtain a detection rate of 98.3% with 0.1% of FPR.

Saxe and Berlin (14) are using manually extracted features from static analysis of executables that they use in a four-layer perceptron model and detect 95% of the malicious files in their dataset at 0.1% FPR. Finally, Raff et al., in Malconv, (12) are using Convolutional Neural Network on the raw

binary file. Their results are a bit inferior to the others, around 90% accuracy, but no preprocessing is required on the data and predictions are done very efficiently.

## 3 DATA SETS

For the experiment, we use 19830 malicious PDF files and 116695 benign ones. The malicious samples are taken from VirusTotal (3). They were uploaded on the website between the 5/5/2018 and the 11/14/2018. They were all detected by at least 10 antiviruses by the 11/20/2018. We also download a set of 9300 newer malicious files, following the same rule, to evaluate the degradation with the time. For this set, the files appeared the website between the 2/15/2019 to the 3/15/2019. The benign files were obtained using collaboration with a private company. Due to privacy issues, we would not be able to make them available. Following the assumptions that the benign PDF evolve much slower than the malicious one, we did not record their date of creation.

We partition our malicious and benign sets into training, validation and test set. In order to make sure our research imitate as well as possible the real-word malware classification challenge, the sets were chronologically organized. For the train set, we use 17880 malicious files, from the oldest one to the ones that appeared in 9/28/2018, and 105027 benign samples. The validation set helps us tuning the hyperparameters of our model and contains 416 malicious files (until 11/10/2018) and 2332 benign. The test set contains the rest.

## 4 NEURAL NETWORK ARCHITECTURE

We design and present two different models. The first one is a minimalist model, with a single convolutional layer, followed by a global max pool and finishing with a linear layer and a sigmoidal gate, we refer to it as ModelA. The second model, ModelB, is similar to ModelA, but has an additional fully connected layer, right after the global max pool. The additional fully connected layer allows the network to combine the filters together and giving him much more freedom, we denote it ModelB. Both the convolutional and fully-connected layers are directly followed by a ReLU in all the networks.

In order to turn the files' byte code to input vectors, we proceed as follows. Initially, to turn them to a fixed size, we select only the first 200kB, if a file is smaller it will be zero-padded at the end. Then, each byte, which is a number between 0 and 255, is mapped to a 16-dimensional vector. To obtain the mapping of the byte $i$, $0 \leq i \leq 255$ we simply compute the product $W^\top x_i$ where $x_i$ is a vector in $\mathbb{R}^{255}$ that is equals to 1 at index $i$ and 0 everywhere else, and $W \in \mathbb{R}^{16 \times 255}$ is the mapping (or embedding) matrix. This operation is actually the first layer of all our networks and the matrix $W$ is trained by back-propagation together with all the other parameters of our models. Such a process is commonly referred to as learning a task-specific embedding, as the matrix $W$ is trained with the network and thus, optimized for our classification problem.

The training has been done for 13 hours on average, on a single Nvidia GPU and we ran 4 epochs on the training data.

As regularization technique, we use early-stopping with the help of our validation set. For ModelB we use dropout with $p = 0.25$ after the fully connected layer.

*The detailed architecture of those two networks can be found in Appendix B.*

Due to its simplicity, ModelA cannot employ complex patterns from the data. It only creates filters for some strings, using the convolutional layer, look for them in the file, and combine them for classification. This is very generic and, in many ways, simulates what an antivirus does with signature matching.
ModelB has an additional fully-connected layer, it is able to combine the output of the filters together. Hence it can create multi-string signatures.

Figure 1

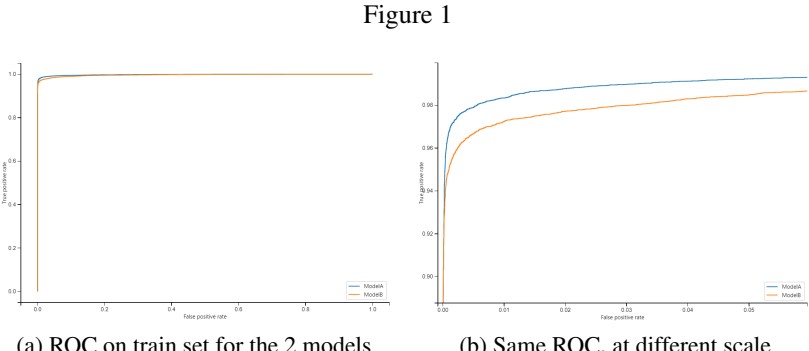

(a) ROC on train set for the 2 models      (b) Same ROC, at different scale

## 5 RESULTS

In this section, we will present the results on our dataset. Following the standard malware detection denomination, we refer to malware as positive examples and benign files as negatives. A False Positive (FP) is a benign file that has been detected as malicious and a True Positive (TP) is a malicious file that has been correctly detected. Similarly, a False Negative (FN) is a malicious file that has been detected as benign and a True Negative (TN) is a benign file that has been correctly detected. We will mainly use two metrics: the detection rate or recall (D) $\frac{TP}{TP+FN}$ and the False Positive Rate (FPR) $\frac{FP}{FP+TN}$. On the first part, we will present the results obtained on the test set, then we will evaluate the degradation of the model on newer files, and we will finish by presenting some examples of (almost) zero-day detection.

Figure 2

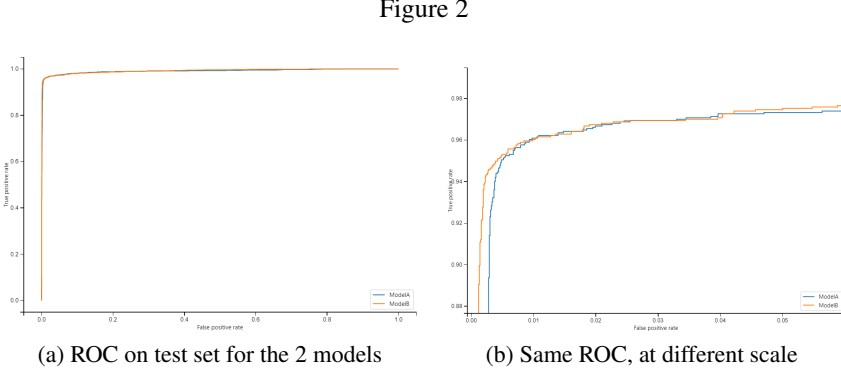

(a) ROC on test set for the 2 models      (b) Same ROC, at different scale

### 5.1 TEST RESULTS

In order to evaluate our algorithm, we compare our results to the antivirus in VirusTotal. As we built our dataset querying for files that have more than 10 detections, a small bias is created towards the antivirus but it still helps us understand how well our algorithm work compared to existing solutions. Due to privacy issues, we decided not to upload the benign files to VirusTotal so we cannot evaluate the False Positive Rate of the vendors. For our models, we will take as reference for the ranking the detection at a 1% FPR but we will present also the detection at 0.5% and 0.2% FPR. The ROCs on train and test displayed in Figures 1 and 2. The results are presented in Table 1.

We can see that our best model ranks in the $7^{th}$ position of all the $52$ antiviruses available on the website at the time of the research. At a $0.2\%$ FPR our models still ranks in the $8^{th}$ position which demonstrate its reliability.

It is worth mentioning here that most antiviruses use hand-crafted signatures and require significant human resources to accomplish this. In our model, no domain knowledge is required, and the signatures are learned automatically by the model.

Table 1: Results on models and some AV on the test set

| Rank | Model | Detection | Detection at 0.5% | Detection at 0.2% |
|---|---|---|---|---|
| 1 | McAfee | 0.993 | | |
| 6 | AVG | 0.975 | | |
| 7 | ModelB | 0.960 | 0.953 | 0.939 |
| 7 | ModelA | 0.961 | 0.950 | 0.807 |
| 8 | GData | 0.940 | | |

Figure 3: Histogram of the AV detection

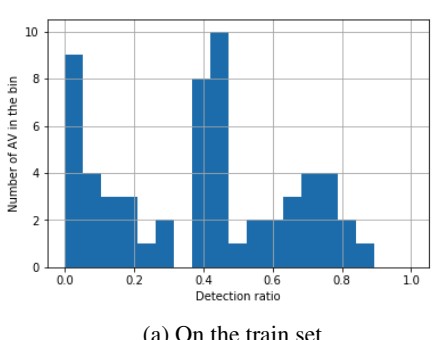

(a) On the train set

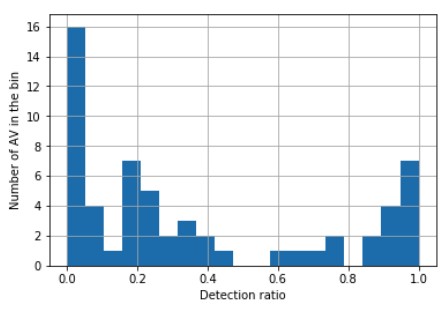

(b) On the test set

We can also note that the ModelB is more robust and efficient than the two other, and maintain a high detection rate even at a FPR of $0.2\%$. Finally we remark that ModelA still performs as well at 1%FPR. This suggests that imitating the AV work, with signature matching, seems like an effective approach for detecting PDF malware.

## 5.2 DEGRADATION WITH TIME

One of the issues with using machine learning to detect malware, is that our detection rate is expected to decrease with time. Indeed, malware are constantly evolving, as hackers keep refreshing their technics, so detection datasets become quickly outdated and models performance degrades on new samples.

In order to evaluate our robustness with time, we download a set of 9300 files from VirusTotal that were seen for the first time between the 15th of February and the 15th March 2019. This makes a 5 months delay from the oldest file of our training set to this new set. In this experiment, our detection for ModelB, using the same threshold of 1% FP, is equal to $97.9\%$ and a bit less for the other ($92.7\%$) . The detection rate for the ModelB got even higher than in the original test set ($96.0\%$). This variation can be explained by the fact that our test set is relatively small and maybe more challenging the new set. In any case, we can observe that our network managed to learn patterns that are generic enough not to suffer from the time gap.

It is interesting to notice that the use of dropout, in the ModelB, increases the chance of generating robust signature. Indeed, running the same network without this operation, does not impact much the detection on the test set ($95.8\%$ at $1\%$ FPR), however, in the new set, it drops to $91.5\%$. This means that proper regularization forces the network to find more generic identifier for malware, and thus, it is less impacted by the time.

## 5.3 ZERO DAYS

With the same goal of investigating generalization capabilities of our model we also look for files that were almost undetected by antiviruses (AV) during our training phase and that started being increasingly detected later. Due to the limitations of the query language in VirusTotal we looked manually for some of those examples.

Figure 4: Histogram of the AV detection

(a) Score distribution for the test set and for the new files

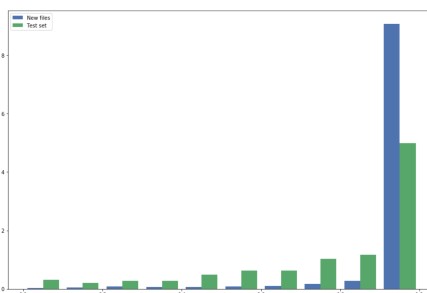

(b) Detection of Sonbokli files two weeks after they were discovered

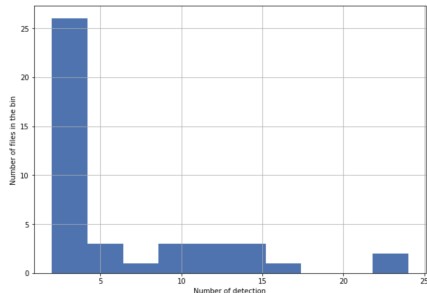

One file we found is labeled as belonging to the Sonbokli family by Microsoft. We downloaded 100 more PDF tagged with this name and having more than 10 antiviruses detecting them in July 2019. Surprisingly, most of these files are presenting the same kind of pattern in Antivirus detection. Initially, a very low number of vendors detect them on the first appearance (usually less than 4, all after January 2019), and, after some times, this number drastically increases, as more vendor detect them.

We run our CNN on those 100 files, which detected 75 of them. This means that our CNN was capable to learn a signature more robust than the one written by most antiviruses for this specific family of malware. This signature was generic enough to still be relevant 8 months after the end of the training period.

Finally, we look for Sonbokli files of the week of the experiment. We found 42 PDF files uploaded to VirusTotal between the first and the 7th of July 2019, none of them had more than 10 detections, and around half of them are detected by Microsoft only. Still, our network is able to detect 81% of them as malicious. We show in Figure 4b the number of antiviruses that detect each file, two weeks after we saw them for the first time.

## 6 Differentiating various families malware

Our ModelA, which has a single convolutional layer followed by a global max pool layer, has essentially the following semantics. The network passes to the max pool layer a list of values indicating the presence of some strings in the file. Those strings, that we will call signature by analogy with the signatures generated by the antivirus vendors, are used to perform the classification. The same signatures are also (potentially) helpful to distinguish different families of malware.

The main issue while trying to classify malware families is that creating their labels is much more complex. The main reason is that antiviruses' names are inconsistent and finding a good heuristic for labelling is a hard problem (8). For that reason, we decide to use unsupervised learning to investigate how well our CNN can differentiate between different malware families. We will use antivirus labels only as a reference for comparison.

In this experiment, we retrieve the 1534 malicious samples of our test set, then run the ModelA network on them while extracting the output of the global max pool layer. This means that for each sample we will be represented by a vector in $\mathbb{R}^{128}$.

### 6.1 Visualization

We first start by displaying in 2D the vectors obtained after the global max pool layer. To do so we will use the common dimensionality reduction technique: T-SNE (7). We would like to compare our results with an antivirus' names. For that, we require the antivirus to detect the major part of our samples while having enough diversity in its labels. The one we found that satisfies the best both of those requirements is Ikarus.

For convenience, we select only the 15 most common families, which drops only 10% of the samples. We also remove the undetected, which represents around 7% of our test. The graph is presented in Figure 5.

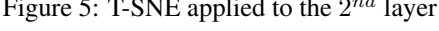

Figure 5: T-SNE applied to the $2^{nd}$ layer

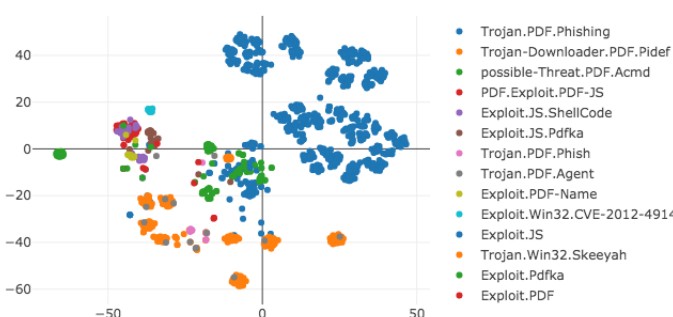

We can easily distinguish the Phishing PDF (Trojan.PDF.Phishing), which is the most common family among the data, from the rest. We can also notice that the Trojan Downloader Pidef family is split among few clusters which seem to be different variations of this malware family. Inside a few of those groups, appear some files labeled 'Agent', meaning that Ikarus could not assign them to a family although our model seems to be able to do so. Another observation is that the families that are based on exploits end up pretty close on the graph, we can guess here that all those malware programs are exploiting the same kind of vulnerability on the system. By looking more specifically at some of these files in VirusTotal, we noticed that most of them are labeled "js-embedded", indicating that they run Javascript on the host computer. This indeed is a strong indicator of maliciousness (6) and explain why we see those groups next to each other.

## 6.2 CLUSTERING

In order to dig deeper into the understanding of the network, we run a clustering algorithm on the vectors described above, and compare the results with three antiviruses: Ikarus, Microsoft and McAfee. We aim to evaluate two metrics: what part of each cluster has the same label (homogeneity) and how much of each family is contained in a specific cluster (completeness).

As a clustering algorithm, we decide to use the density-based algorithm HDBSCAN (10), as it does not require any hyper-parameter except the minimum of files in the cluster, does not make a prior assumption on the shape of the clusters and allows some files not to be grouped. We run the algorithm on the samples without prior filtering, defining a minimum size of 10 files by cluster.

To make the results easily explainable we stick on a simplified definition of homogeneity and completeness.

- The homogeneity is estimated by the ratio between the most common label in the cluster with the number of element in it. More formally, let $L$ be the set of all the names given by a specific antivirus. The homogeneity $H_i$ for cluster $C_i$ is given by

$$H_i = \max_{l \in L} \frac{|\{(c, l) : c \in C_i\}|}{|C_i|}$$

- The completeness is given by the ratio of the biggest family in the cluster that is contain in it. Let $\hat{l}_i = \arg\max_{l \in L} |\{(c, \hat{l}) : c \in C_i\}|$ the most common family in $C_i$. The

completeness $T_i$ of $C_i$ is given by:

$$T_i = \frac{|\{(c, \hat{l}_i) : c \in C_i\}|}{|\cup_j \{(c, \hat{l}_i) : c \in C_j\}|}$$

Note that, for the computation of the homogeneity we do not take into account the files in the clusters that are not detected. The detection rate by cluster and by antivirus can be found in the Appendix A.

Figure 6 is displaying our results by clusters and by antivirus for the two introduced metrics. Out of the 1534 files, only 285 did not end up in any of the 14 created clusters. The most common family in each cluster are listed in the Appendix A.

Figure 6

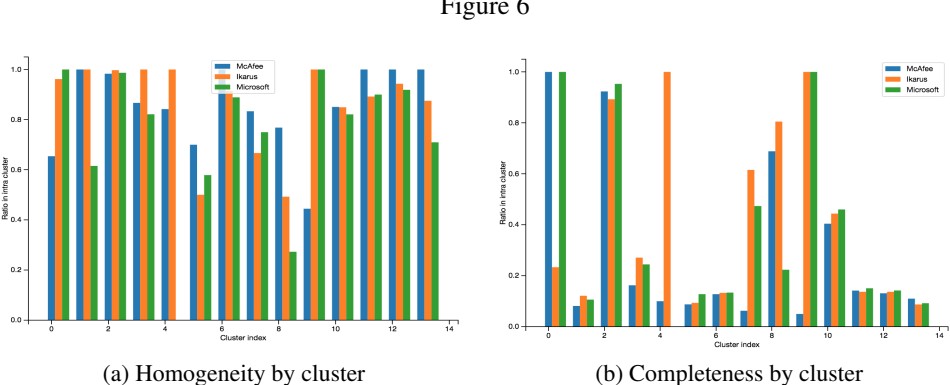

(a) Homogeneity by cluster          (b) Completeness by cluster

First, we can easily notice that, the clusters are much more homogeneous than complete. Indeed we ended up with few clusters for the *Trojan-Downloader Pidef* or for the *possible-Threat Acmd*. This indicates that our algorithm is failing to find a unique identifier for those families and instead is detecting several variants. We also remark that we have two clusters of exploit (7, 8) with a lower level of homogeneity. As suggested in the previous section, our algorithm seems more generic that the AV, and we find few families in each of those clusters without differentiating them. Finally, an interesting observation is the difference between the antiviruses at detecting and differentiating some families. For example, we can notice than in cluster 4, Ikarus labels the files *Skeeyah* but detects only half of them whether McAfee is using the generic name *Artemis* but detects all of them, in this cluster Microsoft does not detect any sample.

## 7 CONCLUSION

We introduced a Convolutional Neural Network to detects malicious PDF using only the byte level of the PDF files. We have shown that our network achieves a performance comparable to some of the best antivirus in the market without requiring any preprocessing or feature extraction. It is able to detect files discovered a few months after its training period and malware that are undetected by most antivirus. Finally, the network proved to be capable of distinguishing accurately various malware families.

Using this kind of approach could save significant time of malware analysts and automate the detection. It will probably be one of the solutions chosen by antivirus companies to cover the drastic increase of new malware files discovered every day.

In future work, we hope to investigate in additional neural network architectures including Recurrent Neural Network (RNN). Additional future research is considering additional meaningful embedding for the raw PDF files, for example, by including some metadata information like tags or xref tables.

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

# A CLUSTERING

## A.1 DETECTION RATE BY CLUSTERS

Figure 7: Detection rate by clusters by AV

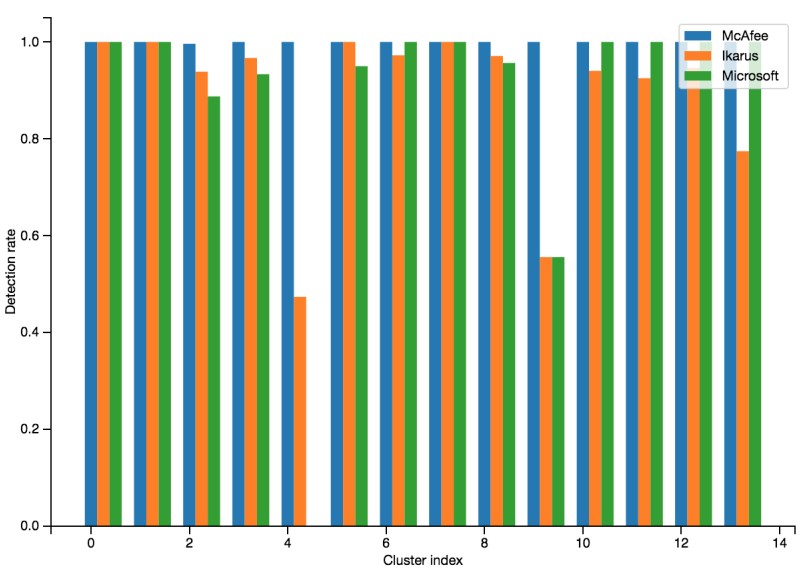

## A.2 CLUSTER'S LABEL

| Cluster | Ikarus | Microsoft | McAfee |
|---------|--------|-----------|--------|
| 0 | possible-Threat.PDF.Acmd | Pdfdrop | Suspicious-PDF.gen.a |
| 1 | possible-Threat.PDF.Acmd | Meterpreter | Artemis |
| 2 | Trojan.PDF.Phishing | Phish | RDN/Generic.dx |
| 3 | possible-Threat.PDF.Acmd | Meterpreter | Artemis |
| 4 | Trojan.Win32.Skeeyah | No detection | Artemis |
| 5 | possible-Threat.PDF.Acmd | Meterpreter | Artemis |
| 6 | Trojan-Downloader.PDF.Pidef | Domepidief | PDF/Phishing.gen.w |
| 7 | Exploit.PDF-Name | ShellCode | Artemis |
| 8 | PDF.Exploit.PDF-JS | Meterpreter | Exploit-PDF.bk.gen |
| 9 | Exploit.Win32.CVE-2012-4914 | CVE-2012-4914 | Artemis |
| 10 | Trojan-Downloader.PDF.Pidef | Domepidief | PDF/Phishing.gen.w |
| 11 | Trojan-Downloader.PDF.Pidef | Domepidief | PDF/Phishing.gen.w |
| 12 | Trojan-Downloader.PDF.Pidef | Domepidief | PDF/Phishing.gen.w |
| 13 | Trojan-Downloader.PDF.Pidef | Domepidief | PDF/Phishing.gen.w |

# B NETWORKS ARCHITECTURES

Figure 8: Model A

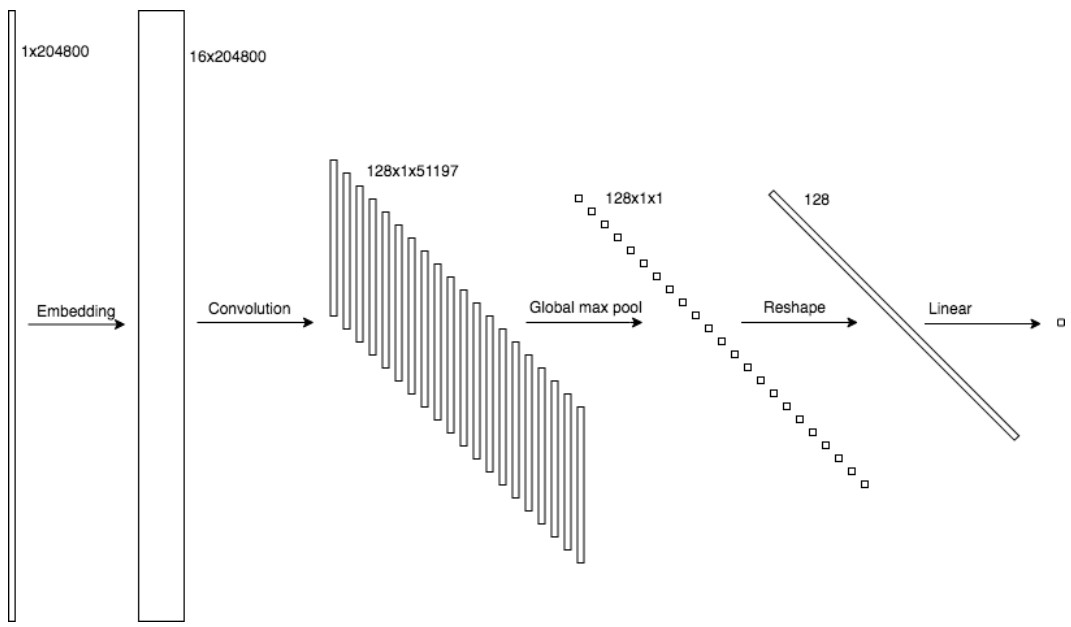

Figure 9: Model B

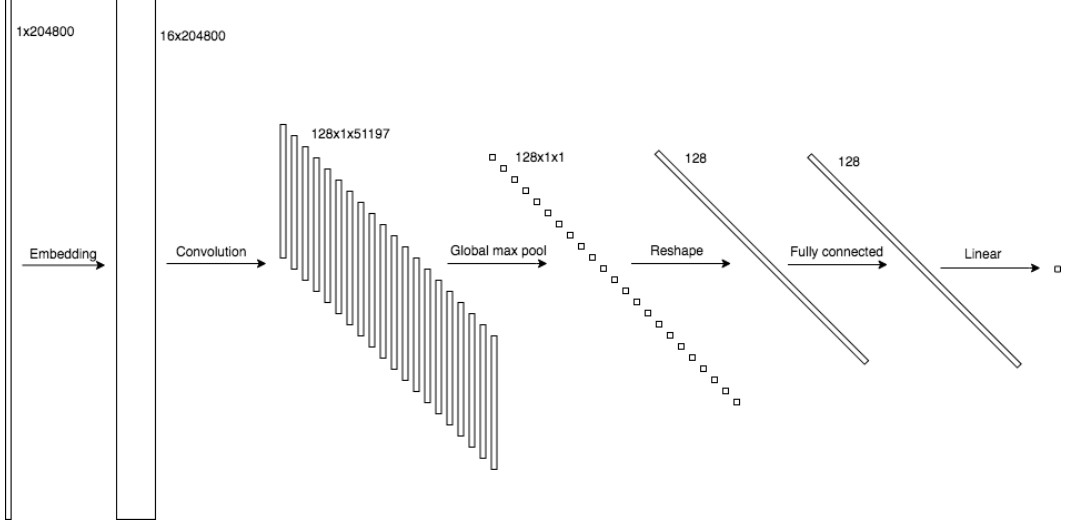

