# OpenReview forum: "Detecting malicious PDF using CNN"
_ICLR.cc/2020/Conference — Reject_

### Official Review · AnonReviewer2 · 2019-10-20
**Official Blind Review #2**

**Rating:** 1

**Review:**

The paper proposes to use CNN to detect malicious PDFs. The paper uses two simple CNN models and train them on a collected dataset, compare with other antivirus software, and conclude that CNN performs better. But I have serious concerns about the experiments.

1. The major concern is with the dataset. The dataset used by this work are collected by authors themselves, and I see a serious problem in this process. The malicious and benign ones are not collected from the same distribution: "The malicious samples are taken from VirusTotal (3). They were uploaded on the website between the 5/5/2018 and the 11/14/2018"; "The benign files were obtained using collaboration with a private company". Thus, the model may not be actually learning what's malicious and what's benign, but only learning whether the pdf comes from that private company or VirusTotal. It's enough since the test set is also collected this way. It can be fairly easy to distinguish between two different datasets, and the same reasoning applies here. Also, the authors are not making the dataset available due to privacy reasons, which further make the dataset's validity a question.

2. Also, antivirus softwares are applicable on any pdf files, but the model trained with the dataset collected may not be useful in other circumstances. The comparison between them are not fair. It should compare the same model/software but on multiple different datasets to demonstrate the model's general applicability.

3. The experiments done in this work is also not of a satisfactory level. For example, there are some missing blanks in Table 1. Why is that? The figures (e.g., Fig 1/2) can be improved a lot. It did not include baseline and provide little information. The fonts are too small.

4. No major novelty is introduced. The work is an application paper using very simple CNNs on the malicious PDF detection problem. This itself does not make the paper bad but combined with the unconvincing experiments it's a serious weakness.

In summary, the paper lacks solid experimental results to make its conclusion convincing and its model generalizable. I vote for rejection of the paper.


**Experience Assessment:**

I have read many papers in this area.

**Review Assessment: Checking Correctness Of Derivations And Theory:**

I carefully checked the derivations and theory.

**Review Assessment: Checking Correctness Of Experiments:**

I carefully checked the experiments.

**Review Assessment: Thoroughness In Paper Reading:**

I read the paper at least twice and used my best judgement in assessing the paper.

---

> ### Author Response · Authors · 2019-11-10
> **Answer to some comments**
>
> Thank for your conscientious review.
> First, we understand your concern about the dataset we used, but there is no standard data set for this task,
> so we had to create our own.
> We decided to focus on recent malware VirusTotal, which was the only place where we managed to find enough samples for training our CNN.
>
> The fact that the t-SNE and the clustering worked well convinced us that our algorithm is actually learning to distinguish different malware families and not the source we got the from.
> We are currently working on bringing data from other sources to further validate this hypothesis.
>
> The blanks in table 1 are justified by the fact that we could not evaluate the antivirus False Positive Rate as, due to the same privacy issue with the benign files, it required us to upload them to VT.

---

### Official Review · AnonReviewer3 · 2019-10-23
**Official Blind Review #3**

**Rating:** 1

**Review:**


The paper proposes using CNNs to detect malicious PDFs by analyzing the bytecode. The authors created a dataset of malicious and benign PDFs to be able to train their model and compared against commercial Antivirus.
The paper

Reject: The paper lacks novelty and have very weak experiments.

The paper presents a simple application of a small CNN to PDF malicious classification, the main contribution of the paper is the dataset collected, which is not released for privacy reasons.

The experiments lack strong baselines, and comparisons with previous ML solutions, how important is the design of the CNN for this task.

Previous work like [12] have already applied the same idea to different but very similar task, malware detection of binary files.

The paper would be more useful for people working in malware detection if made available as a  tech-report.


**Experience Assessment:**

I have published in this field for several years.

**Review Assessment: Checking Correctness Of Derivations And Theory:**

I assessed the sensibility of the derivations and theory.

**Review Assessment: Checking Correctness Of Experiments:**

I assessed the sensibility of the experiments.

**Review Assessment: Thoroughness In Paper Reading:**

I read the paper at least twice and used my best judgement in assessing the paper.

---

### Official Review · AnonReviewer1 · 2019-10-24
**Official Blind Review #1**

**Rating:** 3

**Review:**

This paper presents a model that detects malicious PDF files. The model applies a Convolutional Neural Network (CNN) to analyze the bytes of the input files. The generated features of CNN achieves good clustering results that are consistent with the ground-truth labels.

This paper applies an existing CNN architecture. I do not observe any novelty in terms of modeling. I suspect this paper is not interesting to most members of the ICLR community. Therefore, I do not suggest the acceptance of this paper.

**Experience Assessment:**

I do not know much about this area.

**Review Assessment: Checking Correctness Of Derivations And Theory:**

I assessed the sensibility of the derivations and theory.

**Review Assessment: Checking Correctness Of Experiments:**

I assessed the sensibility of the experiments.

**Review Assessment: Thoroughness In Paper Reading:**

I made a quick assessment of this paper.

---

### Decision · Program_Chairs · 2019-12-19

**Decision:**

Reject

**Comment:**

This submission addresses the problem of detecting malicious PDF files. The proposed solution trains existing CNN architectures on a collected dataset and verifies improved performance over available antivirus software.

There were a number of concerns raised about this work. The main concern the reviewers had with this submission is lack of novelty. The issue is that the paper tackles a standard supervised classification problem which has been extensively explored in the literature and applies an off-the-shelf classification model. Though the particular application has seen less attention in the ICLR community, the problem setting and solution are well known. Thus, the contribution of the work is not sufficient for acceptance.